# Observable Symptoms of Anxiety in Individuals with Fragile X Syndrome: Parent and Caregiver Perspectives

**DOI:** 10.3390/genes13091660

**Published:** 2022-09-16

**Authors:** Reymundo Lozano, Talia Thompson, Jayne Dixon-Weber, Craig A. Erickson, Elizabeth Berry-Kravis, Sara Williams, Elizabeth Smith, Jean A. Frazier, Hilary Rosselot, Cristan Farmer, David Hessl

**Affiliations:** 1Department of Genetics and Genomic Sciences, Icahn School of Medicine at Mount Sinai, New York, NY 10029, USA; 2Department of Pediatrics, Icahn School of Medicine at Mount Sinai, New York, NY 10029, USA; 3Department of Pediatrics, University of Colorado School of Medicine, Aurora, CO 80045, USA; 4National Fragile X Foundation, Washington, DC 20005, USA; 5Department of Pediatrics, Cincinnati Children’s Hospital Medical Center, Cincinnati, OH 45229, USA; 6Department of Pediatrics, Rush University Medical Center, Chicago, IL 60612, USA; 7Department of Psychiatry, Chan Medical School, University of Massachusetts, Worcester, MA 01655, USA; 8Neurodevelopmental and Behavioral Phenotyping Service, National Institute of Mental Health, Bethesda, MD 20892, USA; 9MIND Institute, University of California Davis Medical Center, Sacramento, CA 95817, USA; 10Department of Psychiatry and Behavioral Sciences, Davis School of Medicine, University of California, Sacramento, CA 95817, USA

**Keywords:** *FMR1* gene, intellectual disability, autism, assessment

## Abstract

Caregiver reports, clinical observations, and diagnostic assessments indicate that most individuals with fragile X syndrome experience high levels of chronic anxiety. However, anxiety is a challenging endpoint for outcome measurement in FXS because most individuals cannot reliably report internal emotional or body states. A comprehensive survey of the presence, frequency, and duration of anxiety-related symptoms and questions to elicit open-ended responses was completed by caregivers of 456 individuals with FXS, ages 2–81 years (87 female, 369 male) and 24 female and 2 male FXS self-advocates ages 15–66 years. Caregivers reported classic behavioral indicators of anxiety, such as avoidance, irritability, motor agitation, and physiological symptoms, as well as behavioral features in FXS such as repetitive behavior, aggression, and self-injury. Self-advocate accounts largely paralleled caregiver data. Factor analyses yielded four factors: (1) increased irritability, aggression, and self-injury; (2) increased physical movement, nervous activity, and restlessness; (3) physical and physiological features of anxiety; and (4) internalizing and gastrointestinal symptoms. Caregivers are capable of observing and reporting behaviors that are valid indicators of anxious states that are usually reported in self-report standardized assessments. These results support the development of an anxiety measure for FXS that minimizes problems with rater inference.

## 1. Introduction 

Fragile X syndrome (FXS) is a single gene disorder caused by a large (>200) CGG repeat expansion in the promoter region of the *Fragile X Messenger Ribonucleoprotein 1* (*FMR1*) gene on the X chromosome, which leads to excessive methylation of the gene, transcriptional silencing, and loss or reduced production of the Fragile X Messenger Ribonucleoprotein Protein (FMRP). FXS is the leading inherited cause of intellectual and developmental disability (IDD) and autism spectrum disorder (ASD). High levels of anxiety, stress, and hyperarousal are consistently reported in this population. Furthermore, a high proportion of individuals with FXS meet diagnostic criteria for at least one anxiety disorder [1]. 

Anxiety is an emotion characterized by feelings of tension, worried thoughts, and physical changes such as increased heart rate, abdominal pain or discomfort, and dizziness. The usual assessment and diagnosis of anxiety relies heavily on the patient’s verbal descriptions of symptoms and experiences, as anxiety is understood to be primarily an internal state. However, young children and people with IDD, especially those who are more significantly affected or those with ASD, frequently experience challenges communicating their internal feelings and emotional states; this is often due to impaired language, communication deficits, cognitive disability and/or limited insight [2]. Therefore, the assessment of anxiety in people with IDD often relies on reports from parents and caregivers, including observations of externally expressed behaviors and verbalizations that are believed to be associated with anxiety. Because this assessment approach is less direct, there is often less confidence in the validity and reliability of proxy reports of anxiety in IDD [3]. DSM-based diagnostic interviews with caregivers of persons with FXS, utilizing appropriate adjustments pertaining to intellectual impairment showed that about 80% of individuals with FXS meet criteria for at least one anxiety disorder, most commonly specific and social phobias, but also generalized anxiety disorder, and selective mutism [1,3,4,5]. Often, anxiety in this population is treatment refractory, despite common use of antidepressant and anti-anxiety medications in studied patients. Caregiver behavior rating scales also indicate high rates of social avoidance, anxiety-related behavior, and internalizing symptoms in individuals with FXS [6]. Not surprisingly, caregivers of people with FXS rank anxiety as an especially critical target for effective treatments [7].

Numerous studies have examined the physiological systems and attentional biases which underly stress, anxiety, and arousal reactions in people with FXS. This work documents aberrant hypothalamic-pituitary-adrenal (HPA axis) activity in response to social and cognitive stressors which are linked to internalizing symptoms and social gaze impairment [8,9,10,11,12,13,14]. In addition, exaggerated sympathetic nervous system reactivity, [15,16,17], greater bias in attention to threatening stimuli [18], and dysregulation of the parasympathetic/sympathetic nervous system have been described (elevated heart rate, abnormal regulation of normal beat-to-beat heart rate variability, and abnormal sweating) [19,20,21,22]. Furthermore, hypersensitivity and failure to habituate to direct social gaze has also been described [23,24,25,26]. While not all physiological measurements are anxiety symptom-specific, the enhanced limbic system response plays an important role in the deficits of frontal executive control of social and non-social stimuli. 

Pilot studies of anxiety measures in FXS showed reproducibility and validity at a considerable level [27], even when the anxiety assessment is based on interpretation of the behavior by others rather than by self-report. Results from initial FXS studies in this vein reported that children who are unable to identify and communicate that they worry about general day-to-day events may exhibit more observable behaviors, resembling active and passive avoidance (e.g., arguing, avoiding difficult tasks, staring off) or having specific phobias and compulsions [28]. Thus, it is critical to identify specific, observable behaviors caregivers attribute to anxiety, that affect quality of life and that can be reliably assessed by caregivers. The identification of such behaviors may facilitate more accurate assessment tools, such as in the creation of quality anxiety measures for clinical evaluation and monitoring, as well as, for clinical trials. 

The purpose of the present study is to describe the frequency, severity, and qualitative characteristics of the observable behaviors that occur during periods when individuals with FXS are believed to be experiencing anxiety. The study also describes anxiety-related symptoms in a small subset of individuals with FXS who were able to provide self-report data. 

## 2. Methods

### 2.1. Anxiety Survey

The initial survey questions were created within a focus group of eight professionals (parents, physicians, psychologists) with extensive expertise in FXS and involved in the National Fragile X Foundation (NFXF). The initial draft survey included both structured, forced-choice questions as well as open-ended questions. In addition to questions related to anxiety, the survey included items detailing demographics (e.g., age, sex) of the individual with FXS. Example of questions pertaining to the presence of anxiety were: is the person with FXS, at times, anxious, worried, nervous or afraid; can the person with FXS state with their words that he/she is anxious; behavioral or verbal symptoms, and physical symptoms, reported when the responder believes the person with FXS is anxious, along with frequency, intensity and duration of such symptoms. For respondents identified as having FXS (self-advocate), the same questions were presented with a change to first person text, (e.g., “what do you do or say?” versus “what do you see or hear?”). 

The draft survey was presented to a focus group including parents of individuals with FXS (eight parents of children ranging from 5 to 35 years of age), two people with FXS, and seven medical providers, with the purpose of confirming that the survey was comprehensive, clear, and respectfully worded. The results prompted the inclusion of duration and intensity to be separately rated for the symptoms observed. Subsequently, the final survey was sent via email to three families (two parents, one family of a teenager, one family of an adult with FXS, and one sibling of an individual with FXS) and reviewed extensively by employees of the NFXF who volunteered to pilot the process involved in completing the survey. No further adjustments were requested based on this pilot sample. No identifying information was included in the survey. The final survey can be seen in the Appendix A and included a total of 21 items. 

The newly generated, anonymous survey was administered online through a survey software program. Under the Protection of Human Subjects section 45 CFR 46.101(b)(2) this study was exempted. The NFXF sent a survey link to 10,000+ emails subscribed to receive its general emails. The NFXF also posted a link to the survey on their social media sites, and a link was posted on their website August through December 2019 at the MyFXResearch portal site. Recipients of the email were eligible to participate if they were an individual with FXS or were a family member/caretaker of an individual with FXS. Participants caring for multiple individuals with FXS were eligible to complete the survey once per individual with FXS. While a total of 1414 people clicked on the survey link, only 482 completed the survey. 

### 2.2. Participants

456 caregivers of individuals with FXS between the ages of 2 and 81 years (87 female, 369 male) completed the survey. Among females with FXS, *n* = 4 were 2–5 years, *n* = 17 were 6–12 years, *n* = 19 were 13–17 years, *n* = 30 were 18–29 years, *n* = 16 were 30–49 years and *n* = 1 was >50 years. Among males, *n* = 33 were 2–5 years, *n* = 80 were 6–12 years, *n* = 70 were 13–17 years, *n* = 102 were 18–29 years, *n* = 74 were 30–49 years and *n* = 10 were >50 years. In addition, 28 self-advocates (24 female and 2 male) age 15–66 (Mean 41.8 ± 15.6) completed the survey. 

### 2.3. Analyses

#### 2.3.1. Quantitative Analyses

Given the numerous behavioral or physical features of anxiety on the survey, a descriptive, rather than hypothesis-driven approach was taken. For both caregiver and self-advocate surveys, participants first indicated whether specific symptoms occurred during periods of anxiety. If the respondent answered “yes”, they were asked to make ratings pertaining to frequency, intensity, and duration of the symptoms. Frequency was rated using the following scale: 1 = Rarely when they are anxious, 2 = Sometimes when they are anxious, 3 = About half the time when they are anxious, 4 = Usually when they are anxious, 5 = Always when they are anxious. Duration was rated using the following scale: 1 = Less than I minute, 2 = 1–5 min, 3 = 6–15 min, 4 = 16–60 min, 5 = Greater than 60 min. Intensity was rated as scale of 1–5 (1 = mild, 2= slightly, 3 = moderately, 4 = very much, 5 = extremely). Because of the subjectivity of the intensity ratings the data were not reported, rather focusing on frequency and duration.

We first examined whether age was associated with the presence or frequency of specific behaviors related to anxiety within females and males using Spearman correlations. Given the exploratory nature of the analysis, uncorrected *p*-values are reported and thresholded using *p* < 0.05. Next, the proportion of participants (all females, given limited N; males within each age bin: 2–5, 6–12, 13–17, 18–29, and 30–49 years) responding “yes” was quantified and rank ordered to allow identification of commonly reported symptoms. The same approach was taken for determining the symptoms that occur most frequently and for duration. These rank-ordered variables were graphed and reviewed to identify prominent patterns.

#### 2.3.2. Factor Analysis

Both male and female participants aged 5–49 were included in the factor analysis. This subset of 418 participants was 80% male (*n* = 333) and 20% female (*n* = 83) with less than 1% unknown (*n* = 2 missing sex). The mean age of the sample was 20 years (SD = 10) (quartiles = 12, 18, 27 years). For the measurement invariance analyses, the subgroup was split into participants younger than 18 years (*n* = 196, 80% male, mean age 12 ± 4) and 18+ years (*n* = 222, 79% male, mean age 28 ± 7).

Each item had four response scales: yes/no, intensity, duration, and frequency. For the purpose of exploratory factor analysis, the yes/no scale was used. A response was not required for any of the items, and some respondents indicated “no” by leaving the item blank. To account for this, all non-responses to the yes/no scale were coded “no.”

The goal of this analysis was to explore a candidate factor structure for the development of an instrument to measure observable behaviors related to anxiety in FXS, and to evaluate its measurement invariance across age groups. The first stage was exploratory data analysis to evaluate cell size and interitem correlations. Second, exploratory factor analysis was conducted. Given the binary response scale, the estimation method was weighted least square mean and variance adjusted (WLSMV). Oblique (geomin) rotation was used. Solutions with up to four factors were requested.

Model fit was evaluated using the relative fit indices root mean square error of approximation (RMSEA), comparative fit index (CFI), and the Tucker-Lewis index (TLI). Smaller values of RMSEA and larger values for CFI and TLI Indicate better fit, and commonly used thresholds are 0.06 for RMSEA and 0.95 for CFI and TLI [29]. However, these fit indices are for continuous data and when used in categorical data they are biased [30]. In addition to the fit of the solution, the quality was evaluated by noting the strength and pattern of item loadings. The selected solution was carried forward to the measurement invariance analysis. For this confirmatory method, items were adopted onto a factor if the loading was greater than 0.40.

Measurement invariance was evaluated in three stages. First, the fit of the model in the subgroups independently was assessed, evaluating whether requiring the same factor structure in both groups was associated with poor model fit. Next, the factor loadings were constrained to be equal across groups and assessed how this impacted model fit. Finally, the item thresholds were constrained to equality across groups. Theta parameterization was used because the data were binary. For this reason, the equivalence of residual variance was not tested. The same relative fit indices used for the exploratory analysis were used, but without thresholds, given that typical conventions do not likely apply when using binary data [30].

#### 2.3.3. Qualitative Analysis

Free-text survey responses about behaviors seen, vocalizations heard, and physical symptoms related to anxiety were analyzed using a qualitative content analysis approach. Content analysis is a systematic approach to text analysis, using codes and categorization to label and organize content trends and patterns within a qualitative data set [31]. Responses were uploaded to ATLAS.ti analytic software. A team-based consensus code was developed by a multidisciplinary team with extensive FXS clinical and research expertise including, parents, physicians (pediatricians, geneticist, neurologist, and psychiatrist), and psychologists [32]. During the coding sessions 20% of the responses were discussed, one at a time, until consensus was reached on which code(s) to apply to each piece of text. The process was iterative; codes were collapsed and combined, and responses were recoded as needed to best represent the data. Next, two team members (psychologist and physician) working as primary coders used the codebook to independently co-code an additional 10% of the data. Intercoder agreement was calculated using Krippendorf’s α and a high rate of agreement (α = 0.96) between the coders supported the trustworthiness of the coding structure and the overall study [33]. The remaining results were divided and coded independently between the primary coders. Next, code frequencies for specific emergent observable behaviors and physical symptoms were calculated and proportions were compared by group (male versus female; children versus adolescents/adults) using Pearson’s Chi-square analysis with significance set at *p* < 0.05. Caregiver responses to the open-ended question: “What makes you confident it is anxiety related?” were coded, categorized, and networks were developed to examine relationships and code co-occurrences. Several broad themes were developed to represent overarching ideas about anxiety in FXS, using direct quotes to support the themes. Final themes were discussed and approved by the entire multidisciplinary research team.

## 3. Results

### 3.1. Developmental Patterns of Behaviors Related to Anxiety Associated with Age in Females and Males with FXS

Among females with FXS, there was no significant association between age and occurrence, or frequency of behaviors associated with anxiety. Furthermore, among females, age was not significantly associated with the occurrence of any physical symptoms of anxiety. Given the smaller sample size of females and lack of clear association with age, further analyses for this group were combined across ages.

Among males, increased age was weakly correlated with several types of behavior, including decreased aggression (−0.14, *p* < 0.01), avoidance (−0.17, *p* < 0.01), resistance (−0.35, *p* < 0.001), hyperactivity (−0.16, *p* < 0.01), refusal (−0.15, *p* < 0.01), throwing objects (−0.29, *p* < 0.001), pacing (0.13, *p* < 0.05) and repetitive speech (0.17, *p* < 0.01). There were some correlations between increasing age and less frequent occurrence of rapid heart rate (−0.11, *p* < 0.05) and flushing (−0.13, *p* < 0.05), and with more frequent report of sweating (0.12, *p* < 0.05). Correlations with duration of behaviors and physical symptoms were all positive (older age related to longer duration of symptoms), reaching generally correlations for avoidance (0.19, *p* < 0.01), refusal (0.18, *p* < 0.01), pacing (0.23, *p* < 0.01), resistance (0.25, *p* < 0.001), negative speech (0.12, *p* < 0.05), throwing (0.17, *p* < 0.05), tension (*p* < 0.05), hyperventilation (0.30, *p* < 0.01), shakiness (0.25, *p* < 0.05), sweating (0.19, *p* < 0.05) and vomiting (0.22, *p* < 0.05).

### 3.2. Females with FXS (All Ages)

What Do Caregivers Observe When They Believe the Person with FXS Is Anxious?

Caregivers reported three behaviors in females with FXS at a high rate (>75% of respondents): refusal (e.g., “refuses to do things, gets on floor and will not move or do anything”), changes in facial expression (e.g., “appears angry, worried, or fearful”), and avoidance (e.g., “running away, refusing to go places or remaining in one’s room, hiding, stays off to the side when in a group”). Moderately common behavioral symptoms (40–75% of caregivers) included: repetitive or negative speech, freezing, aggression, and fidgetiness. Less common symptoms (<40%) included self-injury, hyperactivity, throwing objects and pacing (Figure 1a). The frequency with which these specific behaviors occurred (ranging from 1, rarely to 5, almost always) followed a relatively similar pattern, except that repetitive speech was most frequent (mean = 3.68), followed by change in facial expression (3.62), and refusal (3.37); however, all behaviors, on average, were seen at least “sometimes” and none averaged “rarely” (Figure 1b).

Duration of behaviors (ranging from 1, <1 min to 5, >60 min) was highest for repetitive speech, avoidance, and refusal (on average 6–15 min), although all behaviors averaged at least 1–5 min in duration. Throwing, pacing and self-injury were of relatively short duration (1–5 min) when they occurred. The most common physical symptoms of anxiety rated by caregivers were, in order: body tension, flushing, rapid heartbeat, and stomachaches reported in 30% to 45% of females (Figure 2). Of note, caregivers reported that 55% of females use words to communicate their anxiety.

### 3.3. Males with FXS

What Do Caregivers Observe When They Believe the Person with FXS is Anxious?

In males, the rate of some specific caregiver-reported behaviors associated with anxiety were dependent on age, while others were consistent across development. Hyperactivity or other increased physical activity was very common among younger males (~80% of 2–5-year-old children), and somewhat less common among school age children and adults (~50–60%). Throwing objects was also very common in young children (~75%) and dropped off with age to approximately 30% by late adulthood (18–29, 30–49 years). Refusal (80–90%), aggression (60–75%), avoidance (55–85%), and self-injury (40–60%) remained relatively common across the age groups (Figure 3).

Caregivers endorsed some behaviors occur at higher frequency when the person with FXS is anxious, including repetitive speech (on average “usually”; and most frequent among males > 6 years), refusal (about half of the time), changes in facial expression and avoidance (both on average between “half the time” and “sometimes”) and pacing (ranging from “sometimes” to “about half time time”) (Figure 4).

Behaviors such as avoidance, repetitive speech, pacing (especially in older individuals), refusal, and hyperactivity tended to be of longer duration compared to aggressive behaviors (throwing, hitting, self-injury). Notably, pacing was the least endorsed symptom in 2–5-year-olds (shortest duration) and the most endorsed (longest duration) in 18–29-year-olds (Figure 5).

As seen with females, physical symptoms of body tension and flushing were the most commonly reported by caregivers of males with FXS. However, gastrointestinal symptoms (e.g., diarrhea in all age groups and vomiting notably in 2–5-year-old children) were more common in males (Figure 2). For males with FXS, caregivers reported ability to express anxiety with words differed by age group, with the following percentages of caregivers reporting “yes”: (2–5 years, 3%; 6–12 years, 19%; 13–17 years, 26%; 18–29 years, 32%; 30–49 years, 36%).

### 3.4. Qualitative Findings

Content analysis of caregivers’ free-text responses revealed several additional behavioral and physical symptoms of anxiety for people with FXS. Most notably, 76/456 respondents (17%) listed crying behaviors associated with anxiety (male = 55/369, female = 21/87). Anxious crying was more common in females than males with FXS (*p* < 0.01), and children (ages 2–12 years) showed significantly more crying than adolescents and adults (ages 13+ years) among both females (*p* < 0.05) and males (*p* < 0.001). Other specific emergent behaviors noted in free-text responses included hand biting (*n* = 37), chewing objects (*n* = 33), hair twirling (*n* = 15), skin picking (*n* = 10), and hand wringing (*n* = 8). Novel physical symptoms for anxiety included various somatic complaints (*n* = 23), grimacing (*n* = 22), and toileting problems (incontinence/encopresis) (*n* = 17).

Content analysis of caregiver responses to the question, “What makes you confident it is anxiety?” resulted in three overarching themes with nine sub-categories (see Table 1 for representative quotes).

*It Looks and Feels Like Anxiety*. Caregivers reported their child’s behavioral and physical symptoms were consistent with anxiety. Some parents simply stated they knew their own child and were especially empathetic based on their own experiences with anxiety. Others described individuals with FXS who appeared scared, fearful, or terrified. Finally, many reported that behaviors and physical symptoms associated with anxiety were a major change from typical behavior for the person with FXS.

*Triggers Imply Anxiety.* For many, the specific triggers made it clear that the individual with FXS had anxiety. Changes in routines and unpredictable settings were reported as particularly triggering for many individuals with FXS. Caregivers also described anticipation of undesirable activities, as well as common fears inducing stimuli such as dogs, heights, crowds, and speaking in public.

*Reasons for Relief Imply Anxiety.* Some caregivers noted that reasons for relief signified the person with FXS had been experiencing anxiety. For a subset of respondents, anxiety treatments (medications) reduced symptoms. However, more frequently caregivers reported reassurance and removal of demands resolved symptoms and provided evidence for anxiety.

### 3.5. Fragile X Syndrome Self-Advocates

#### 3.5.1. Quantitative Findings

What do self-advocates experience when they are anxious? Among the 26 self-advocates, the most commonly reported features of anxiety were rapid heartbeat (81%), followed by change in facial expression (69%), avoidance (65%), nervous activity (62%), fidgeting (62%), followed by “zoning out”, refusal, and shakiness, body tension and freezing (each at 58%). Self-advocates rated the frequency with which specific behaviors or experiences occurred (ranging from 1, rarely to 5, almost always). On average, features such as shakiness, sweating, body tension, fidgeting, and hyperactivity most often accompanied anxiety, however the pattern was quite heterogeneous, with many different features occurring “usually” or “almost always” with anxiety (e.g., nearly all features except throwing, vomiting, hives, and hiccups which occurred in a small proportion of self-advocates).

#### 3.5.2. Qualitative Findings

Among the 26 self-advocates, all (100%) reported experiencing anxiety, worry, or fear, and 19 (73%) reported that they are able to verbalize to others that they have these experiences. Respondents wrote that they use words such as “I am anxious and I need to take a few deep breaths”, “Terrified, scared, anxious”, “I am so anxious I am freaking out”, “Worried and afraid about almost anything”, and “I am worried about…”. One respondent wrote: “Mainly my anxiety is at a 10 right now.”

### 3.6. Exploratory Factor Analysis Results

Exploratory Data Analysis. Cell size was <25% for six items: hiccups (*n* = 18, 4%), hives (*n* = 26, 6%), shakiness (*n* = 72, 17%), and stomachache (*n* = 102, 24%), and high endorsement for repetitive speech (*n* = 345, 83%) and refusal (*n* = 360, 86%). Although the proportion was low, the absolute number of participants was sufficient, so these items were retained for analysis. The tetrachoric correlation matrix for all items indicated weak associations among most variables; 75% of correlations had a magnitude smaller than 0.25.

Exploratory Factor Analysis. The fit indices for the solutions with one through four factors are shown in Table 2. All fit indices improved with additional factors. The three- and four-factor solutions both had acceptable numbers of items with strong loadings on each factor, although the three-factor solution had more cross-loading (Table 2).

Measurement Invariance Analyses. Both the three- and four-factor solutions were reasonable candidates, and so they were carried forward to the measurement invariance stage. We adopted items which had an absolute loading of >0.40 onto a factor. Cross-loading did not occur, with one exception: Zoning was included on both factors 1 and 3 of the three-factor solution, even though the loading on factor 3 did not exceed 0.40 (it was 0.38). This is because the first stage of measurement invariance analysis indicated poor fit unless the item was included on both factors. Table 3 shows the fit indices for each stage of invariance testing for the three- and four-factor solutions, respectively. These results indicate that fit does worsen with increasing model constraints, but the changes are minimal, and the fit indices stay in the same range. These results are generally supportive of the invariance of both the three- and four-factor solutions across age.

Interpretation of the Four Factor Model. Factor 1 includes relatively high loading items related to aggression, irritability, and self-injury. Factor 2’s higher loading items include increased physical movement, nervous activity and restlessness. Factor 3 includes higher loading items tapping physical and physiological features of anxiety such as rapid heart rate, hyperventilation, physical shakiness and tension, and changes in facial expression. Finally, Factor 4’s emphasis appears associated with gastrointestinal distress and internalizing symptoms (Table 4).

## 4. Discussion

Caregivers of people with FXS identify anxiety as the most problematic symptom for which the highest number of caregivers would like a drug developed [7]. Yet, when existing measures of anxiety have been discussed with the Food Drug Administration (FDA), they were not acceptable as a primary endpoint because the participant with FXS cannot self-report and the caregiver must make some assumptions about internal feeling states or worrisome thoughts to complete many rating scale measures. In order to develop an anxiety measure that can be used in clinical trials, we must therefore identify the observable behaviors reported by caregivers that are felt to represent anxiety [34,35,36].

Our results reflect patterns of behaviors and physical features encompassing the body of clinical experience regarding manifestation of anxiety in patients with FXS. Many symptoms reported by caregivers and self-advocates, both in quantitative data and qualitative responses, fall in line with those captured by classic DSM criteria for anxiety disorders. These include increased avoidance, somatic and GI symptoms, enhanced sympathetic nervous system activation, and psychomotor agitation. Others are very commonly associated with anxious or fearful states in FXS but do not necessarily fall under the purview of the DSM. These include repetitive speech, aggression, refusal, and hyperactivity (in younger individuals, pacing in older). These latter features can be present in persons with FXS when they are not, or are less anxious, but they seem to be exacerbated during periods of anxiety and increased stress. Also, although externalizing symptoms and aggression such as throwing objects, threatening behaviors, and refusal may not on the surface appear to be anxiety symptoms per se, it is important to note that irritability and agitation are symptoms of anxiety, and numerous clinician and caregiver reports suggest links between anxious states and irritability or aggression. Our findings indicate that behaviors are very similar to those reported in a prior study of behaviors that parents and teachers identified as anxiety-related in children with FXS, including confrontational and non-confrontational avoidance, repetitive acts or compulsions, and various forms of behavioral dysregulation (nervous movements, skin picking, talking excessively, tics) [28].

As stated earlier, one of the main criticisms of caregiver-reported symptoms of anxiety in people with FXS is that an external rater cannot directly observe or legitimately infer internal emotional states or thoughts of the patient (e.g., cannot “see” worry, fear, or anxious feeling states). The present factor analysis result appears to provide preliminary evidence that caregivers can record observations of behaviors related to anxiety that reliably cluster into categories of symptoms that reflect the clinical understanding of anxiety in this population. In other words, caregiver accounts of these behaviors are not randomly occurring, disassociated behaviors, but rather co-occur in ways that are consistent with the clinical understanding and manifestation of anxiety in people with and without FXS. Although the ratings included in the factor analysis had limitations (e.g., dichotomous yes/no), the four-factor solution nevertheless provided evidence that certain types of similar symptoms (e.g., aggression, self-injury; signs related to physiological changes) tend to cluster together within and across individuals.

Qualitative themes identified from free-text responses to the question, “What makes you confident the behaviors are anxiety related?” provided additional evidence for the reliability of caregiver reporting. Caregiver respondents spontaneously chronicled triggers that are commonly associated with anxiety (e.g., changes in routines, anticipation of events, exposure to common phobia stimuli). Furthermore, parents described well established treatments for anxiety as reducing their child’s observable behaviors, including SSRI medications, relational supports, and behavioral strategies. Qualitative content analysis also revealed some behaviors we did not anticipate, including crying as a symptom of anxiety in younger children and females. These combined data suggest parents may actually be uniquely suited to observe and identify anxiety in individuals with FXS, provided rating forms or clinician interviews are structured to elicit accurate reports. The high levels of supervision required for many individuals with FXS coupled with empathetic caregiving relationships can allow parents to notice anxious states through minor or major shifts in facial expressions and behavioral patterns, all within the child’s natural home environment and daily routines.

The quantitative and qualitative information related to anxiety provided by FXS self-advocates closely parallels data provided by caregivers, providing additional evidence that specific and observable behaviors can reasonably, though certainly not always, be linked to anxious or nervous internal states. These parallels are especially notable for aspects of avoidance, nervous motor movements, and refusal behaviors. These first-hand accounts of anxiety in individuals with FXS are extremely rare in the literature and provide an invaluable patient-centered perspective that may to some extent generalize to experiences of others with FXS who may not be as able to express their feelings or symptoms. These data also suggest that some individuals with FXS, who are able to communicate their insights, are able to contribute directly to the development of an anxiety measure by serving as patient-advocates in a consulting/advisory role to ensure that patient experiences and challenges with anxiety are well-represented.

Although females with FXS have on average significantly higher cognitive and verbal abilities than males with FXS, the spectrum of anxiety-related behaviors was quite similar overall in both sexes. Comparably, refusal and avoidance were highly endorsed symptoms in both sexes. In males, aggression and pacing were also highly endorsed and more prominent symptoms than in females. Gastrointestinal symptoms such as vomiting appeared more often in males as a manifestation of anxiety. These findings are consistent with clinical experience with patterns of behavior in males versus females. Age effects were not strong for individual behaviors although there were some patterns that emerged, such as non-specific hyperactivity in younger boys with FXS evolving to pacing in young adults. The frequency of overall observed behaviors declined substantially with age although duration increased for some behaviors, supporting a pattern seen in clinical care-more frequent less severe behaviors in young children and infrequent but more severe behaviors in adults.

The limited insight and communication inability to express worries or complaints along with somatic symptoms of anxiety may lead to behavioral outbursts in FXS. Participants reported that tantrums and resistance to changes in the routines or environment in individuals with FXS were often preceded by anxious, nervous, or fearful states. It is possible that repetitive speech, often repetitive questioning to seeking constant reassurance, and related vigilance to insist on sameness may express worry about changes in the near future. Perseverance, routines, overreaction to environmental changes, social avoidance, and withdrawal seem to reflect anxiety as reported by the participants. These observations may explain the positive correlations of the Aberrant Behavior Checklist (ABC) Irritability and Lethargy/Social Withdrawal subscales observed in previous studies [17,27,37,38]. The ABC Irritability and Lethargy/Social Withdrawal subscales which have been used as outcome measures in clinical trials may be surrogates of anxiety, but they do not capture the full spectrum of anxiety manifestations in FXS. Thus, these individual ABC subscales may not always provide adequate sensitivity to successfully show an intervention is operating on anxiety in FXS, leading to false negative results for a treatment that is potentially effective. Further, because the two ABC subscales may capture different expressions of anxiety in FXS, both would be required to capture more complete effects on anxiety across the population, and models for using such a composite measure in FXS have not yet been accepted by regulatory authorities. Since current anxiety surrogate measurements may not provide adequate sensitivity to the entire range of behaviors felt to represent anxiety in FXS, a new measure is needed to improve the likelihood of successful trials targeted to this phenotype in FXS.

The factor analysis identifies categories of behavior that group together and can be assessed as domains in a rating measure tapping behavioral manifestations of anxiety in FXS. The analysis results will be critical to the design of such a measure, which could be a 3- or 4-domain clinical global impression scale and an overall “observable behaviors indicating anxiety” rating with the rating for each domain incorporating and anchored based on numbers of behaviors exhibited in the domains, and taking into consideration frequency, duration and severity of the behaviors identified. Such a measure will need to be evaluated for clinician understanding of the concepts being rated, test-retest reproducibility and inter-rater reliability, convergent validity with other caregiver-reported, and clinical interview-based measures of anxiety and aberrant behavior (e.g., the Pediatric Anxiety Rating Scale [PARS] [27]; the ABC-FXS scoring algorithm [39]; and the Anxiety, Depression and Mood Scale [ADAMS] [1,40]. It will be important to obtain caregiver feedback on clinical meaningfulness of changes in rating, and to relate potential severity of anxiety presentation to quality of life. A validated measure also would have utility in many treatment settings that may involve myriad interventions including therapy, pharmaceuticals, genetic rescue, or potential combinations therein, among other approaches to alleviate symptom burden in FXS.

The limitations of the study include that the survey was lengthy and complex, requiring responses regarding several features of symptoms. There is also the possibility of recall bias. While the sample size of self-advocates was small, these reports are extremely valuable, as they provide first-hand accounts with which to compare caregiver ratings. There are also limitations of the analysis, particularly in the exploratory factor analysis, which were addressed in the methodology and results section.

## 5. Conclusions

The present study was an in-depth analysis of a comprehensive survey of symptoms and behaviors associated with anxiety, nervousness, and fear in individuals with FXS, a group with very high rates of anxiety disorders. The data collected from caregivers and self-advocates, both quantitative and qualitative, indicates that people with FXS experience both classic symptoms of DSM-defined anxiety disorders and a number of additional symptoms or reactions that appear to be generated by anxious states. While most people with FXS cannot reliably describe their internal body or emotional states to support self-reported assessment procedures, the data presented here provides evidence that caregivers can reliably identify behaviors that are meaningful for anxiety. These behaviors may be represented on outcome measures that can be developed, validated, and later employed in clinical and research settings.

This work was made possible by partnering with families affected by FXS and self-advocates. The qualitative accounts in particular provide family and patient voices to enhance the ecological validity and lived experience of the anxiety symptoms. This should ensure generalizability and patient representation in future outcome measures development. Partnering with individuals and their families living with FXS is important for all research projects, and particularly to ensure that concepts being measured in new treatment development are those that families and caregivers have identified as important to them for quality of life Therefore, involving patient advocacy groups, convening family advisory committees, and implementing patient feedback into the design and execution of research is imperative to its success. The National Fragile X Foundation and other patient advocacy groups have programming to support researchers in gathering and implementing community feedback that can inform meaningful endpoint development for regulatory approval of new treatments.

## Figures and Tables

**Figure 1 genes-13-01660-f001:**
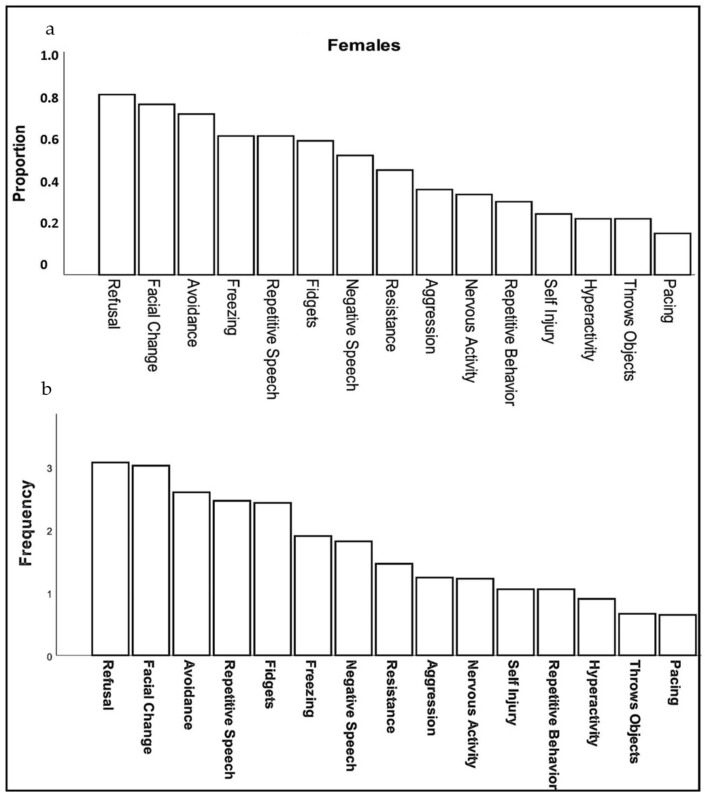
Proportion (**a**) and frequency (**b**) of females with FXS reported by caregivers to show specific behaviors or verbal symptoms when they believe the individual is experiencing anxiety.

**Figure 2 genes-13-01660-f002:**
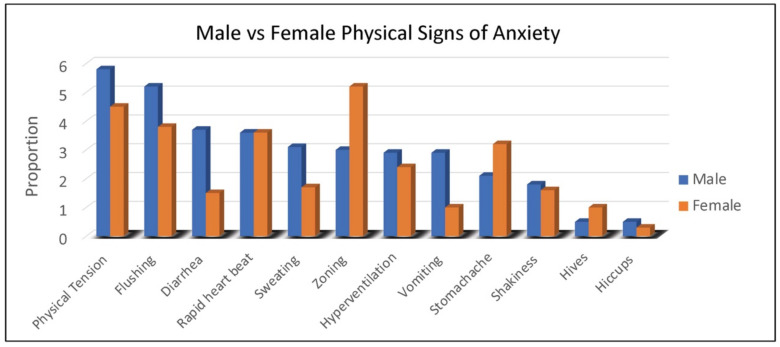
Proportion of males and females with FXS reported by caregivers to experience specific physical symptoms when they believe the individual is experiencing anxiety.

**Figure 3 genes-13-01660-f003:**
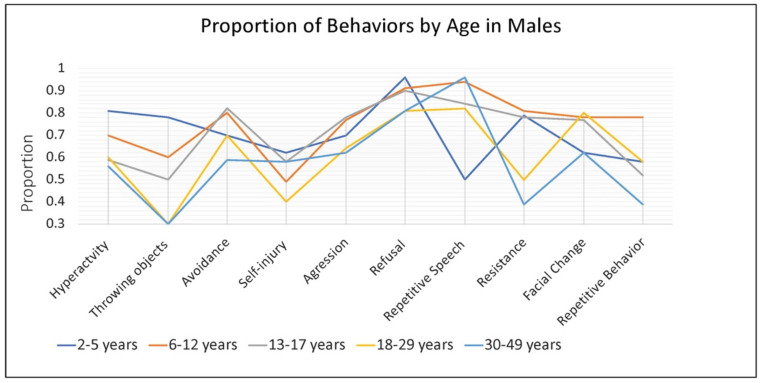
Proportion of males with FXS, within each age group, reported by caregivers to show specific behaviors or verbal symptoms when they believe the individual is experiencing anxiety.

**Figure 4 genes-13-01660-f004:**
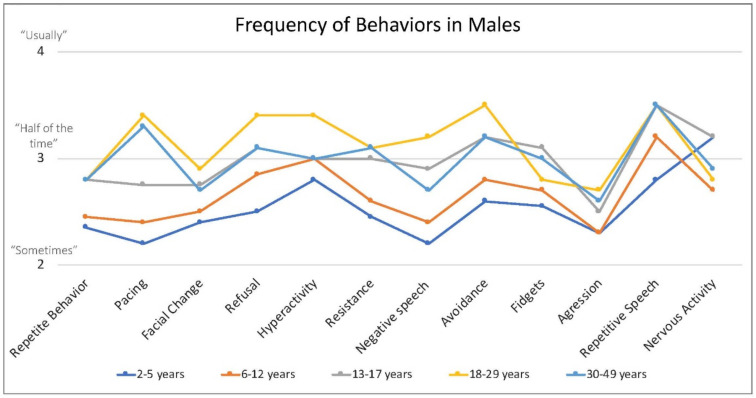
Frequency of behaviors by age in males with FXS reported by caregivers. 1 = rarely when they are anxious, 2 = sometimes when they are anxious, 3 = about half the time when they are anxious, 4 = usually when they are anxious, 5 = always when they are anxious.

**Figure 5 genes-13-01660-f005:**
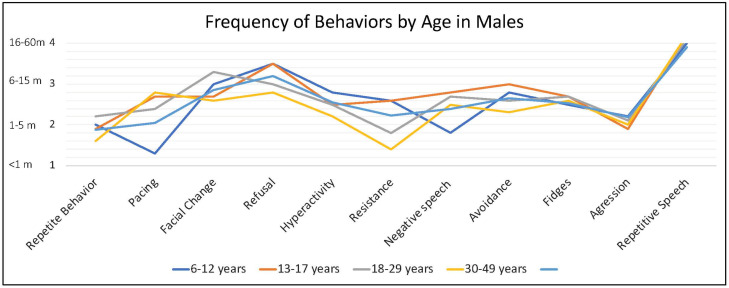
Duration of caregiver-reported specific behaviors and verbal symptoms associated with anxiety in males with FXS. 1 = less than 1 min, 2 = 1–5 min, 3 = 6–15 min, 4 = 16–60 min.

**Table 1 genes-13-01660-t001:** Qualitative analyses: Themes, categories, and quotes.

It Looks and Feels Like Anxiety		
I Know My Child and Myself	Fearful Appearance	Change from Normal
“I’m her mother/caregiver. I have known her for 24 yrs. I know all the signs just by looking at her face.” “I endure similar symptoms and I am mindful of my own signs.” “I’m the mother and I just feel. He is anxious. That’s what I see.” “The signs are always the same and because some similar issues when my anxiety kicks in.”	“He strikes out, you see fear in his eyes.” “He is visibly upset and scared, it’s hard to watch because of the fear of the unknown on his face, sometimes the fear becomes anger, and he lashes out verbally and yells, chews on his right pointer finger.” “You can hear the sheer panic or distress in our son’s voice & actions.”	“It’s different than when he usually talks to himself or repeats because he is not happy when he is doing it. It looks more like he is scared of something and it’s harder to talk to him or console him.” “I only see these behaviors when he is anxious. If I see or hear him alone, talking to himself, I know he is anxious about something.” “Change in attitude to defensive or ‘frozen.’”
**Triggers imply anxiety**		
Change in routine	Anticipation of undesirable activities	Fear inducing stimuli
“If my son sees someone in the supermarket like (his teacher) he becomes shy and very anxious. In his mind she’s out of context. She belongs at school, not at the market.” “It usually involves new situations, something not in the repetitive normal schedule. Different surroundings, something never experienced before. Fear of the unknown.”	“It is usually before something, even if not something new as long as it is something he does not prefer or mentally or physically more challenging then he likes it to be.” “The behaviors appear at a time when anxiety would typically present. For example, when I present an unwanted activity like going to school.”	“Generally, it is related to a known scary presence like a dog.” “The symptoms appear to be more extreme reactions to situations that would cause a typically developing person anxiety (e.g., moving to college, starting a summer job at camp, unexpectedly being asked to make a menu choice in an unfamiliar place).”
**Reasons for relief imply anxiety**		
Anxiety treatment works	Reassurance	Removal of demand
“Having lived with him for 14 years and noting his difference on and off Zoloft he definitely has anxiety. It’s fairly well controlled most days.” “Starting Zoloft or increasing the dose ameliorates the various behaviors.”	“If I hug/provide comfort, tell him I love him, tell him that he’s ok, his body relaxes.” “He can be soothed to some degree by a familiar, calm person. He responds favorably to reassurances, though you need to stay with him for a while or it will start back.”	“It ends when whatever causes it stops; i.e., when he meets the demand successfully or the demand is removed.” “Changing the subject, getting through the situation, reminders of what will come next or removing the situation calms or extinguishes the behaviors.”

**Table 2 genes-13-01660-t002:** Exploratory factor analysis model fit indices.

Factors	Parms	Chi Square	DF	CFI	TLI	RMSEA [95% CI]	SRMR
1	26	619.740	299	0.745	0.723	0.051 [0.045, 0.056]	0.118
2	51	470.578	274	0.844	0.815	0.041 [0.035, 0.048]	0.100
3	75	365.283	250	0.909	0.881	0.033 [0.026, 0.04]	0.087
4	98	299.924	227	0.942	0.917	0.028 [0.018, 0.036]	0.078

**Table 3 genes-13-01660-t003:** Measurement invariance analyses model fit indices.

	DF	Chi-Square	CFI	RMSEA
**Three-Factor**				
CFA in Young Subgroup	185	251.0314	0.839	0.043
CFA in Old Subgroup	185	255.3515	0.884	0.041
Step 1 (identical structure)	370	506.0962	0.866	0.042
Step 2 (identical loadings)	389	518.5372	0.873	0.040
Step 3 (identical thresholds)	386	529.088	0.859	0.042
**Four-Factor**				
CFA in Young Subgroup	113	150.4461	0.892	0.041
CFA in Old Subgroup	113	118.9891	0.985	0.015
Step 1 (identical structure)	226	269.4738	0.943	0.030
Step 2 (identical loadings)	239	277.0592	0.950	0.028
Step 3 (identical thresholds)	235	287.5166	0.931	0.033

**Table 4 genes-13-01660-t004:** A 4-factor solution results of factor analysis of caregiver-reported behaviors, verbal symptoms and physical symptoms associated with anxiety among 418 individuals with FXS, showing the most significant loadings (orange color) of items onto factors.

4-Factor Solution				
Observed Behavior	Factor 1	Factor 2	Factor 3	Factor 4
Aggression	0.82	0.04	−0.01	−0.03
Avoidance	0.01	0	0.49	−0.1
Facial change	0.08	0.01	0.54	−0.1
Fidgets	−0.06	0.38	0.11	0.04
Freezing	−0.26	0.01	0.36	0.05
Hyperactivity	0.14	0.69	0	−0.05
Nervous Act	−0.02	0.36	−0.01	0.2
Refusal	0.31	0.03	0.21	0.03
Pacing	0	0.72	−0.03	−0.05
Repetitive Behavior	0.27	0.37	0.2	0
Self-injury	0.51	−0.02	0.01	0.14
Negative Speech	0.67	0.07	0.08	0.05
Repetitive Speech	0.17	0.28	−0.03	0.13
Throws Objects	0.62	0.02	0.08	−0.08
Diarrhea	0.21	0	−0.02	0.67
Flushing	0.15	0.22	0.16	0.13
Hiccups	−0.09	0.36	−0.01	0.29
Hives	0.07	0.06	0.05	0.07
Rapid Heart Rate	0.11	−0.04	0.65	−0.06
Tension	−0.02	0.05	0.62	−0.13
Hyperventilation	0.13	−0.02	0.74	0.07
Shakiness	−0.02	−0.1	0.55	0.12
Stomachache	−0.13	0.02	0.3	0.55
Sweating	0.01	0.18	0.41	0.25
Vomiting	0.27	−0.06	0.01	0.51
Zoning	−0.42	0.22	0.31	0.02

## Data Availability

All data were kept confidential and was stored securely with password protected access enabled.

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
