# Peer review of "Observable Symptoms of Anxiety in Individuals with Fragile X Syndrome: Parent and Caregiver Perspectives"

_genes, 2022, doi:10.3390/genes13091660_

Round 1

Reviewer 1 Report

Line 56 – ‘and’ seems to be missing between intellectual and developmental or they can represent it as ‘intellectual/developmental disability

Line 213 – based on the description of the qualitative analysis employed it resembles content analysis more than thematic analysis, so it is not clear why the authors mention thematic analysis in line 214. Would have been good to clearly describe and integrate content analysis description in their mode of usage – so they have described how they used it, but would be good if they can integrate the process of the analytical method. So far it is not clear if they used content or thematic analysis, seems more like the first.

Some information, briefly in the methods section re ethical process/approval would have been good.

Figure one seems a little blurry, could it be made better in terms of resolution?

Table 4 seems more to one side?

Overall, the paper presents an interesting and important findings in relation to IDD/ASD and mental health – such findings can contribute and inform resolutions and measures to improve the quality of life of individuals with IDD/ASD.  

Author Response

Reviewer 1. We appreciate the reviewers' comments and provide a point-by-point response below. 

Line 56 – ‘and’ seems to be missing between intellectual and developmental or they can represent it as ‘intellectual/developmental disability. 

We added the word “and” 

Line 213 – based on the description of the qualitative analysis employed it resembles content analysis more than thematic analysis, so it is not clear why the authors mention thematic analysis in line 214. Would have been good to clearly describe and integrate content analysis description in their mode of usage – so they have described how they used it, but would be good if they can integrate the process of the analytical method. So far it is not clear if they used content or thematic analysis, seems more like the first. 

We appreciate the suggestion to remove the term thematic analysis from the methods, as the process was much more closely aligned with a qualitative content analysis approach. We have restructured the methods to better explain the process of qualitative content analysis in general (with an added citation) and have delineated the ways in which we applied content analysis to 1) identify emergent observable behaviors and physical symptoms of anxiety not previously identified through closed-ended questions, and 2) develop themes about how and why parents ascribe these behaviors to anxiety 

Some information, briefly in the methods section re ethical process/approval would have been good. 

We have included this information in lines 256-258 

Figure one seems a little blurry, could it be made better in terms of resolution? 

We have improved the resolution of figure one.  

Table 4 seems more to one side? 

This was corrected 

Reviewer 2 Report

Is is possible to add more information of

1. The percentage of FXS males and femaled with evident anxiety disorder or autism spectrum disorder?

2. severity/level of intellectual disability

3. comorbidity (state of health)

Author Response

Reviewer 2. We appreciate the reviewers' comments and provide a point-by-point response below. 

Is it possible to add more information of 

  1. The percentage of FXS males and females with evident anxiety disorder or autism spectrum disorder?

All participants (100%) express the presence of anxiety (line 518) and the evident physical signs are reported in figure 2 for both sexes. Unfortunately, we did not collect information about comorbidities, including state of health, ASD or level of IDD.  

  1. severity/level of intellectual disability? As above
  2. comorbidity (state of health)? As above